# The Assessment of Muscle Mass and Function in Patients with Long-Standing Rheumatoid Arthritis

**DOI:** 10.3390/jcm10163458

**Published:** 2021-08-04

**Authors:** Hye-Won Yun, Chun-Ja Kim, Ji-Won Kim, Hyoun-Ah Kim, Chang-Hee Suh, Ju-Yang Jung

**Affiliations:** 1Department of Nursing, Andong Science College of Nursing, 189 Seoseon-gil, Seohu-myeon, Andong 36616, Korea; dntntn@naver.com; 2College of Nursing and Research Institute of Nursing Science, Ajou University, 206 Worldcup-ro, Yeongtong-gu, Suwon 16499, Korea; ckimha@aumc.ac.kr; 3Department of Rheumatology, Department of Nursing, Ajou University School of Medicine, 164 Worldcup-ro, Yeongtong-gu, Suwon 16499, Korea; jwk722@naver.com (J.-W.K.); nakhada@naver.com (H.-A.K.); chusuh@ajou.com (C.-H.S.)

**Keywords:** sarcopenia, arthritis, rheumatoid, glucocorticoids

## Abstract

Muscular dysfunction in rheumatoid arthritis (RA) can affect the quality of life and comorbidities. We enrolled 320 patients with RA, and evaluated their muscle mass, grip strength, and physical performance. Seven (2.2%) and 21 RA patients (6.6%) had sarcopenia, as defined by the European and Asian Working Group for Sarcopenia (EWGS and AWGS), respectively; 54 patients (16.9%) were determined to have low muscle mass with normal muscle function, as defined by the EWGS; 38 patients (11.9%) reported sarcopenia by SARC-F questionnaire. Male sex (odds ratio (OR) 140.65), low body mass index (BMI) (OR 0.41), and use of tumor necrosis factor (TNF) inhibitors (OR 4.84) were associated with a low muscle mass as defined by the EWGS, while male sex, old age, and low BMI were associated with sarcopenia as defined by the AWGS. Old age (OR 1.11), high BMI (OR 1.13), and a high Disease Activity Score 28 (OR 1.95) were associated with sarcopenia as reported on the SARC-F. Male, low BMI, and use of TNF inhibitors were associated with a low muscle mass, while male sex, old age, and low BMI were associated with sarcopenia in patients with long-standing RA.

## 1. Introduction

Rheumatoid arthritis (RA) is a chronic inflammatory disease that causes progressive joint destruction, reducing quality of life, and leading to various comorbidities, including musculoskeletal dysfunction and cardiovascular diseases [1]. The prevalence of RA is known to be approximately 0.5–1% worldwide, and 44% and 16% of RA patients having mild to moderate and severe joint dysfunction, respectively [2]. The development of novel therapeutic agents and the establishment of treatment guidelines for RA have enhanced the ability to prevent the joint damage and deformity; however, morbidity from this disease and its medication are increasing [3]. As a substantial proportion of patients with RA are diagnosed during old age and the proportion of elderly individuals in the population is growing, the number of elderly patients with RA is expanding, and comorbidities related to aging are increasing [4,5].

In RA, loss of muscle and change of body composition occur through a variety of complicated mechanisms, including decreased physical activity, chronic inflammation, drugs affecting patients’ diet and body metabolism, and mood disturbance [6,7]. Patients with RA also commonly have sarcopenia, which is characterized by a generalized loss of skeletal muscle mass and function and an increased risk of fragility [8,9]. Sarcopenia is known to be a risk factor for osteoporosis and fracture, decreasing the quality of life and immune function of patients, while increasing the risk of cardiovascular disease and the metabolic syndrome [10,11,12,13]. In certain populations, including elderly and those with cancer, cardiovascular diseases, and liver disease, a reduced lean mass has been associated with mortality, with a hazard ratio of 1.2–1.5 [13]. As the proportion of elderly RA patients increases, it will become increasingly important to diagnose and prevent sarcopenia in this population.

In 2010, the European Working Group on Sarcopenia in Older People (EWGSOP) defined criteria for the diagnosis of sarcopenia, including evaluations of muscle mass strength, and physical performance, while the Asian Working Group for Sarcopenia (AWGS) modified these criteria based on Asian studies [8,14]. As described in these guidelines, sarcopenia involves a quantitative decrease in muscle function. For patients with clinically suspected sarcopenia, the SARC-F (sluggishness, assistance in walking, rising from a chair, climb stairs, falls) questionnaire was developed to quickly screen for sarcopenia based on the cardinal features or consequences of this disease [15,16]. SARC-F has been considered valid and consistent for identifying people at a risk of sarcopenia associated adverse outcomes, and had a consistent reliability and validity in the African American Health study, Baltimore Longitudinal Study of Aging, and National Health and Nutrition Examination Survey. SARC-F scores of ≥4 are associated with poor muscle function, hospitalization, and mortality. To date, this tool has never been evaluated in patients with RA.

In this study, we compared different groups identified by the sarcopenia classification criteria of the EWGS, AWGS and SARC-F, and determined the clinical features associated with low muscle mass or strength in Korean patients with RA.

## 2. Materials and Methods

### 2.1. Study Participants

All subjects met the 1987 revised criteria of the American College of Rheumatology or the 2010 American College of Rheumatology/European League Against Rheumatism Classification criteria for RA. All of them were 18 years of age or older and less than 80 years old, diagnosed RA within 10 years, and had no history of malignancy and other autoimmune or inflammatory disorders. Subjects participated in this study from April 2020 to December 2020 at the outpatient clinic of Ajou University of medical center. Data related to medical history, symptoms, and physical examination findings, laboratory results were collected through chart review and interview, and entered into a database. Patients were excluded if they refused to provide consent for participation.

### 2.2. Measurement of Body Composition and Muscle Mass Index

We collected data about various subject characteristics, including height, weight, and body mass index (BMI). Height was measured using a height meter (HIE-401^®^; Hanilsporex, Pocheon, Korea), and weight and BMI were measured using a body composition analyzer (Inbody 770^®^; Biospace, Seoul, Korea). The muscle mass index was also measured with a body composition analyzer (Inbody IOI353^®^; Accuniq, Daejeon, Korea) using a touch-type electrical stimulation method and the criteria for sarcopenia recommended by the AWGS (2019). These criteria defined a reduction in muscle mass as a value of (skeletal muscle mass)/(height)^2^ of less than 5.4 kg/m^2^ for women. Muscle strength was measured using a grip dynamometer (TKK5401^®^; Takei Scientific Instruments, Tokyo, Japan), as described by Mentzel et al. [17]. In brief, the muscle strength was measured while encouraging maximum force for 4–5 s, and after 60 s of rest, the measurements was repeated, alternating right and left in the same manner. The higher value among the two measurements was used. If the grip strength was less than 18 kg, it indicated muscle loss. Physical performance was determined by measuring the time (seconds) it took subjects to walk 4m at a usual walking speed in accordance with the Asia’s Muscle Reduction Diagnosis Criteria (AWGS, 2019) for women. If this duration was less than 0.8 m/s (4 m, more than 7 s), it corresponded to muscle reduction. Appendicular lean mass index (aLM) was defined as the appendicular skeletal muscle mass/height^2^ (kg/m^2^).

In this study, three different sets of criteria for sarcopenia were used, the EWGS, AWGS, and SARC-F. Each set of criteria utilized different cutoff values for low muscle mass, strength, and performance. For the AWGS, a low muscle mass was defined by an aLM of ≤7.0 kg/m^2^ in men and ≤5.7 kg/m^2^ in women, while an aLM ≤8.87 kg/m^2^ in men and ≤6.42 kg/m^2^ in women in the EWGS [8,9]. Having low muscle mass without an impairment on muscle strength or physical performance was defined as “presarcopenia”, and “sarcopenia” is defined as low muscle mass with low muscle strength or low physical performance. The SARC-F is a screening questionnaire, consisting of five questions, pertaining to muscle strength, walking aids, getting up from a chair, climbing stairs, and falling [15,16]. This tool utilized a 3-point scale as follows: “not difficult at all” (0 points), “a little difficult” (1 point), or “very difficult to perform” (2 points). If the total score was 4 or more, it indicated sarcopenia, with a Cronbach’s α of 0.81.

### 2.3. Statistical Analysis

The subjects’ general characteristics, disease and drug-related characteristics, and factors related to muscle reduction were analyzed by using descriptive statistics including frequency, error, percentage, mean, and standard deviation. Difference in the indices of muscle reduction were analyzed using the Fisher’s exact test, χ^2^ test, *t*-test, multiple logistic regression for groups with or without sarcopenia, and changes in body fat distribution were analyzed using an independent *t*-test and Pearson’s correlation. For reliability, the Cronbach’s alpha coefficient and test–retest reliability coefficient were calculated to verify the internal consistency and stability of the tool. Logistic regression analysis was performed for factors related to muscle reduction, such as drug and steroid doses. Data analysis was performed using SPSS 23.0 (SPSS Inc., Chicago, IL, USA), and a two-sided statistical significance level of *p* < 0.05 was utilized.

### 2.4. Ethics Statement

This study was conducted in compliance with the principles of the Declaration of Helsinki. The Medical Ethics Committee of Ajou University Hospital Institutional Review Board approved the study protocol (IRB No. AJIRB-BMR-SUR-20-053). All patients agreed to participate in this study and provided written informed consent.

## 3. Results

### 3.1. Clinical and Sarcopenic Characteristics of RA Patients

Table 1 summarizes the clinical characteristics of 320 patients with RA. The mean age was 60.5 ± 9.8 years, and 20 (6.6%) were male. The mean duration of RA was 104.5 ± 67.5 months, the mean Disease Activity Score 28 (DAS28) was 3.0 ± 1.1, and 78 patients (24.1%) had bony erosions due to RA. Nineteen patients (5.9%) were defined as having sarcopenia as defined by the EWGS, and 54 patients (16.9%) had low muscle mass without low muscle strength or physical performance (presarcopenia) as defined by the EWGS (Appendix A). Additionally, 7 patients (2.2%) were defined as having sarcopenia as defined by the AWGS, 38 patients (11.9%) were reported sarcopenia by SARC-F questionnaire. The mean strength of grip was 19.3 ± 5.8 kg, and the mean walk speed was 0.8 ± 0.2 m/s.

### 3.2. Comparison of Clinical Factors between RA Patients with Low Muscle Mass and Those Without

When comparing patients with or without a low muscle mass as defined by the EWGS, those with a low muscle mass had a higher proportion of male patients (20.4% vs. 3.4%, *p* < 0.001), and a lower weight (48.7 ± 8.6 kg vs. 59.6 ± 9.3 kg, *p* < 0.001), and BMI (19.7 ± 2.8 kg/m^2^ vs. 24.3 ± 3.3 kg/m^2^, *p* < 0.001) (Table 2). In addition, more patients with a low muscle mass had bony erosions (37.0% vs. 21.4%, *p* = 0.015) and a higher current (1.8 ± 1.3 mg vs. 1.3 ± 1.3 mg, *p* = 0.011) or cumulative (5.3 ± 6.0 g vs. 3.3 ± 3.6 g, *p* = 0.002) dose of glucocorticoids. Moreover, patients with a low muscle mass had a higher a prevalence of osteoporosis (63% vs. 40.2%, *p* = 0.002) and a lower total mass index (23.5 ± 11.2 kg/m^2^ vs. 27.6 ± 11.0 kg/m^2^, *p* < 0.001) and fat free mass index (12.6 ± 4.3 kg/m^2^ vs. 19.2 ± 5.4 kg/m^2^, *p* < 0.001).

### 3.3. Comparison of Clinical Factors between RA Patients with Sarcopenia by Definition of SARC-F

When comparing patients with or without sarcopenia as defined by the SARC-F, patients with sarcopenia had a higher age (67.3 ± 8.7 years vs. 59.6 ± 9.7, *p* < 0.001), and BMI (24.9 ± 3.5 kg/m^2^ vs. 23.3 ± 3.7 kg/m^2^, *p* = 0.007) but a lower height (152 ± 6.8 cm vs. 157.4 ± 6.1 cm, *p* < 0.001) (Table 3). Patients with sarcopenia also had a higher visual analogue score (35.3 ± 21.3 vs. 26.0 ± 16.4 mg, *p* = 0.015), tender joint count (7.0 ± 7.0 vs. 3.5 ± 3.5 mg, *p* = 0.001), swollen joint count (2.9 ± 4.6 vs. 0.9 ± 1.8, *p* < 0.001), and DAS28 (3.8 ± 1.4 vs. 2.9 ± 1.1 mg, *p* < 0.001). Moreover, the current dose of glucocorticoids was higher (1.4 ± 1.3 mg vs. 1.2 ± 1.0 mg, *p* = 0.011), while the cumulative dose of glucocorticoids was lower (2.8 ± 3.0 g vs. 3.7 ± 4.2 g, *p* = 0.002) in patients defined as sarcopenia by SARC-F. Additionally, the total mass index (22.2 ± 9.9 kg/m^2^ vs. 27.5 ± 11.2 kg/m^2^, *p* = 0.005) were lower, while the fat free mass index (19.7 ± 5.3 kg/m^2^ vs. 17.9 ± 5.8 kg/m^2^, *p* = 0.032) and the aLM (7.6 ± 1.3 kg/m^2^ vs. 7.3 ± 1.1 kg/m^2^, *p* = 0.036) were higher in patients defined as sarcopenia by SARC-F. Finally, the walk speed (0.78 ± 0.16 m/s vs. 0.84 ± 0.16 m/s, *p* = 0.035) and mean hand grip strength (15.4 ± 5.4 kg vs. 19.8 ± 5.6 kg, *p* < 0.001) were lower in patients with sarcopenia as defined by the SARC-F than in those without sarcopenia.

### 3.4. Associations of Clinical Factors with Low Muscle Mass or Sarcopenia in RA Patients

Using a multiple regression analysis, male sex (odds ratio (OR) 140.65, *p* < 0.001), BMI (OR 0.41, *p* < 0.001), and use of tumor necrosis factor (TNF) inhibitors (OR 4.84, *p* = 0.037) were associated with a low muscle mass in RA patients, while the presence of erosion, DAS28, cumulative dose of glucocorticoids, and osteoporosis were not (Table 4). Sarcopenia as defined by the EWGS was associated with BMI (OR 0.66, *p* < 0.001), and sarcopenia by the AWGS was associated with male sex (OR 41.03, *p* = 0.003), age (OR 1.19, *p* = 0.038), and BMI (OR 0.47, *p* = 0.002) in RA patients. Sarcopenia as defined by the SARC-F was associated with age (OR 1.11, *p* < 0.001), BMI (OR 1.13, *p* = 0.015), and DAS28 (OR 1.95, *p* < 0.001) in RA patients.

### 3.5. Comparison of Muscular Parameters by Disease Activity and Joint Damage

To compare muscle mass and function between patients with or without active disease, patients with RA were divided into groups with DAS28 either lower or higher than 3.2. These groups did not differ in their aLM (7.37 ± 1.05 vs. 7.26 ± 1.15 kg/m^2^, *p* = 0.255) nor walk speed (0.82 ± 0.16 vs. 0.85 ± 0.17 s/4m, *p* = 0.081); however, the group with a DAS28 of ≥3.2 had a lower mean grip strength (17.77 ± 5.9 vs. 20.53 ± 5.36 kg, *p* < 0.001) than the group with a DAS28 of <3.2 (Figure 1).

To compare muscle mass and function between patients with or without joint damage, patients with RA were also subdivided according to the presence or absence of bony erosion. The aLM (7.25 ± 1.35 vs. 7.32 ± 1.02 kg/m^2^, *p* = 0.079) and walk speed (0.82 ± 0.17 vs. 0.84 ± 0.16 s/4m, *p* = 0.565) did not significantly differ between these two groups; however, the group with bone erosion had a lower mean grip strength (17.27 ± 5.46 vs. 19.95 ± 5.72 kg, *p* = 0.001) than those without erosion.

## 4. Discussion

In our sample of 320 patients with long-standing RA, 2.2–6.6% patients were found to have sarcopenia, while 16.9% had presarcopenia, which is a condition of low muscle mass without compromised muscle strength or physical performance [9]. Patients with a low muscle mass were found to have lower body weight and BMI and were taking higher doses of glucocorticoids than those with a normal muscle mass. In addition, the proportion of males, presence of bony erosion, and use of TNF inhibitors were higher in patients with a low muscle mass. While a low muscle mass was associated with the preceding factors, only a low BMI was found to be a risk factor for sarcopenia as defined by the EWGS. On the other hand, male sex, old age, and low BMI were determined to be risk factors for sarcopenia as defined by the AWGS. Patients with sarcopenia defined by the SARC-F were found to be older, had a higher BMI, and had higher visual analogue score for arthralgia, higher tender joint counts and swollen joint counts, and higher DAS28. Although patients with sarcopenia as defined by the SARC-F were found to be taking higher current doses of glucocorticoids, they were also found to have taken lower cumulative doses of glucocorticoids. Old age, high BMI and disease activity were found to be risk factors for sarcopenia as defined by the SARC-F.

Rheumatoid cachexia or sarcopenia has been reported in 4.5–54.8% of patients with RA, with the prevalence varying depending on the instrument or criteria used for diagnosis, age, disease activity, and the ethics of the study population [18,19]. The definition of a reduced muscle mass differs across these criteria, and the occurrence of sarcopenia depends not only on the disease-related factors, but also on nutrition, physical activity with occupational feature, and socioeconomic status of the study populations. In a study of 457 Chinese patients with RA, 45% had myopenia, and these patients had a higher disease activity, functional limitation index, and radiographic joint damage score [20]. In a study of 388 female Japanese RA patients aged 54.3–72.0 years old with an average disease duration of 9 years and a DAS28 of 2.0–3.5, 37.1% were classified as having sarcopenia, with 49% having a low muscle mass, representing a much higher proportion than the 16.9% identified in our study population [21]. In the previous Chinese study, patients were older (49.5 vs. 60.5 years), had a longer disease duration (54 vs. 104.5 months), and had less use of biologic agents (6.6 vs. 33.1%) than in our population. The patients in both studies had joint damage, including erosion and Steinbrocker staging; however, this damage was more common in the previous studies than in our population. While the age, disease status, and medication profiles were comparable, the proportion of patients with bone erosion was less in the present study, suggesting the disease activity had been well-controlled for a longer period of time. However, our study’s cross-sectional design only involved sampling of disease activity at one time point, limiting its ability to characterize the disease burden, including the inflammatory status and physical function, over a long period of time.

Several studies have shown that joint damage and high disease activity are risk factors for low muscle mass in RA [6,20,22,23]. A systematic review with meta-regression analysis showed that disease duration, DAS28, Health Assessment Questionnaire (HAQ) score were associated with occurrence of sarcopenia in RA [24]. Studies using experimental murine models of arthritis have revealed a loss of muscle mass derived from active arthritis [25,26]. However, the current study showed that the muscle mass was not associated with the DAS28 or bone erosion. Therefore, sarcopenia in RA patients is not explained by disease activity alone, and various other factors must contribute to this phenomenon, including systemic inflammation, which is known to be involved in metabolic processes like muscle generation and degradation and can cause muscle loss.

While a low BMI was found to be associated with a low muscle mass and muscle dysfunction, the DAS28 and bone erosion were not associated with muscular parameters. Obesity or overweight has been found to negatively influence control of disease activity and the quality of life in patients with RA [27,28]. In contrast, a higher BMI has been associated with a lower probability of radiographic progression in RA [29,30]. These previous studies were conducted in patients with early RA (≤12 months); however, our patients had long-standing RA (104.5 ± 67.5 months). In the present study, a low BMI was not associated with disease activity nor damage but was a risk for sarcopenia, which is associated with falls and fractures.

TNF inhibitors are frequently used in RA patients with moderate-to-high disease activity if conventional disease-modifying antirheumatic drugs (DMARDs) fail to control disease activity. TNF-α have been shown to lead to pathophysiological decrease in muscle mass and induce skeletal muscle protein loss and apoptosis in myoblast and myofibers through TNF-α/nuclear factor (NF)-κB signaling [31,32,33]. The effects of TNF-α on muscle protein synthesis are concentration-dependent, with high levels inducing proliferation and low levels stimulating differentiation after muscle damage or aging [34,35]. Infliximab, an anti-TNF agent, has been shown to reverse muscle volume and strength in patients with Crohn’s disease; however, an anti-TNF intervention did not block muscle atrophy in an arthritis animal model [36,37]. In the present study, use of anti-TNF agents was associated with a low muscle mass in RA patients, and the role of anti-TNF agent in muscle wasting should therefore be assessed prospectively in RA patients in future studies. The use of anti-TNF agent may indicate long-term, high, or sustained disease activity that is not represented by the DAS28 at a specific time point or the presence of damage. In that respect, our findings may demonstrate the long-lasting burden of disease leading to sarcopenia.

The SARC-F questionnaire was developed to screen for sarcopenia and is composed of five components: strength, assistance in walking, rising from a chair, climbing stairs, and falls, with a sensitivity of 14% and specificity of 93% compared with the AWGS [38,39]. The SARC-F questionnaire can indicate muscular dysfunction, including low muscle strength and poor physical performance; however, it cannot detect changes in muscle mass. In the EWGSOP algorithm, sarcopenia is confirmed by a measurement of muscle quantity and quality after positive results are found on the SARC-F questionnaire and low levels of muscle strength are identified [9,40]. In the present study, low muscle mass was not correlated with sarcopenia on the SARC-F, though patients with sarcopenia on the SARC-F had a higher aLM than those without sarcopenia. Therefore, the actual muscle mass may not be represented by a patient’s perception of his or her muscular status on the SARC-F, and muscle loss in patients with RA should be identified using other instruments that can better determine muscle quantity.

Though previous studies have shown that the usage or dosage of glucocorticoids can contribute to the development of sarcopenia, the cumulative dose of glucocorticoids was not associated with muscular parameters after adjustment in our study. This finding may have resulted from changes in the pattern of glucocorticoid use in RA patients. The current guidelines, which recommend switching between different kinds of biologic agents, have made it unnecessary to depend on glucocorticoid use to suppress the inflammatory response or to achieve low disease activity or remission [41]. Indeed, most patients in our study were taking a low dose of glucocorticoids (1.4 ± 1.3 mg/day) at baseline. Though the development of sarcopenia is affected by glucocorticoid use, it cannot be concluded that sarcopenia is caused by the use of glucocorticoids in RA patients.

This study had some limitations. The influence of clinical features, including disease-related factors and medication patterns, is limited because the data were collected cross-sectionally, and several factors, including socioeconomic status and nutrition, were not included. In addition, the number of male patients was small compared to the general proportion of male patients with RA. It is considered that there were many cases in which male patients were reluctant to admit that they lacked muscle or muscle strength even in the hospital or were reluctant to provide sufficient time for joining investigations. Although all subjects were enrolled equally according to the established criteria, there might be a selection bias in this study. Muscular parameters were measured by the bioelectrical impedance analysis (BIA) method, which is considered less accurate than the use of other methods like computed tomography, magnetic resonance imaging, and dual energy X-ray absorptiometry. However, these data did not reveal factors associated with the development of low muscle mass or sarcopenia in RA patients being treated with low-dose glucocorticoids, DMARDs, various biologics, or targeted therapy following a recently changed strategy.

In conclusion, 2.2–6.6% of our study population were found to have sarcopenia, with 16.9% having a low muscle mass. A low muscle mass was associated with male sex, low BMI, and TNF inhibitor use, and sarcopenia was associated with a low BMI. Hand grip strength was lower in patients with moderate-to-severe disease activity on the DAS28 and in patients with erosion, while muscle mass and walking speed did not differ between groups divided by disease activity or joint damage.

## Figures and Tables

**Figure 1 jcm-10-03458-f001:**
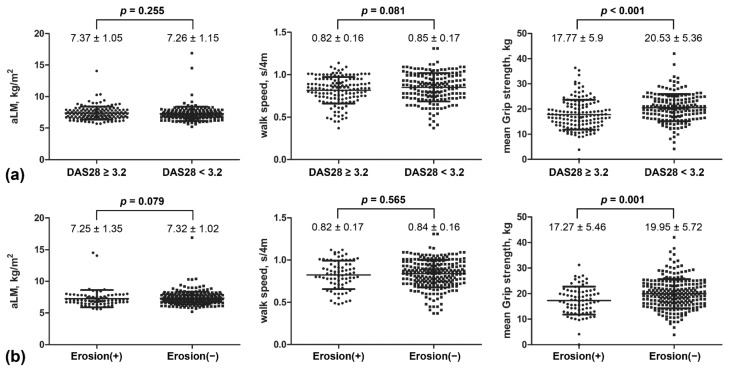
(**a**) Comparisons of appendicular lean mass index (aLM), walk speed, and grip strength in rheumatoid arthritis patients with a disease activity score 28 (DAS28) of ≥3.2 and those with a DAS28 of <3.2. (**b**) Comparison of aLM, walk speed, and grip strength in rheumatoid arthritis patients with bone erosion and those without.

**Table 1 jcm-10-03458-t001:** Baseline characteristics of patients with RA (*n* = 320).

	Mean ± SD or *n* (%)
Age, years	60.5 ± 9.8
Male, *n* (%)	20 (6.6)
Postmenopause, *n* (%)	259 (86.3)
Body weight, kg	57.8 ± 10.0
Height, cm	156.7 ± 6.4
BMI, kg/m^2^	23.5 ± 3.7
Smoker, *n* (%)	21 (6.0)
Duration of RA, months	104.5 ± 67.5
Hypertension, *n* (%)	85 (26.6)
Dyslipidemia, *n* (%)	104 (32.5)
Diabetes, *n* (%)	26 (8.1)
Tender joint count	3.9 ± 4.2
Swollen joint count	1.1 ± 2.4
DAS28	3.0 ± 1.1
Erosion, *n* (%)	77 (24.1)
Visual analogue score	6.4 ± 2.1
ESR, mm/hr	15.7 ± 15.1
CRP, mg/dL	0.8 ± 1.6
Positive rheumatoid factor, *n* (%)	244 (76.3)
Medication history	
Current dose of glucocorticoids, mg	1.4 ± 1.3
Cumulative dose of glucocorticoids, g	3.6 ± 4.1
MTX, *n* (%)	192 (59.8)
Hydroxychloroquine, *n* (%)	163 (50.8)
Leflunomide, *n* (%)	39 (12.2)
TNF inhibitor, *n* (%)	23 (7.2)
JAK inhibitor, *n* (%)	45 (14.0)
NSAID use, *n* (%)	240 (74.8)
Osteoporosis, *n* (%)	141 (44.1)
Total muscle mass, kg/m^2^	26.87 ± 11.1
Fat free mass index, kg/m^2^	18.1 ± 5.8
aLM, kg/m^2^	7.3 ± 1.1
Mean strength of grip, kg	19.3 ± 5.8
Walk speed, m/s	0.8 ± 0.2
Presarcopenia by the EWGS, *n* (%)	54 (16.9)
Sarcopenia by the EWGS, *n* (%)	21 (6.6)
Sarcopenia by the AWGS, *n* (%)	7 (2.2)
Sarcopenia by the SARC-F, *n* (%)	38 (11.9)

RA, rheumatoid arthritis; SD, standard deviation; BMI, body mass index; DAS28, Disease Activity Score 28; ESR, erythrocyte sedimentation rate; CRP, C-reactive protein; MTX, methotrexate; TNF, tumor necrosis factor; JAK, Janus kinase; NSAID, non-steroid anti-inflammatory drugs; aLM, appendicular lean mass index; EWGS, European Working Group for Sarcopenia; AWGS, Asian Working Group for Sarcopenia; SARC-F, sluggishness, assistance in walking, rising from a chair, climb stairs, falls. Values are mean ± SD or n (%).

**Table 2 jcm-10-03458-t002:** Comparison of clinical characteristics between RA patients with low muscle mass and those without.

	Without Low Muscle Mass *n* = 266	With Low Muscle Mass * *n* = 54	*p*-Value
Age, year	60.1 ± 9.7	62.4 ± 10.3	0.061
Male, *n* (%)	9 (3.4)	11 (20.4)	<0.001
Height, cm	156.7 ± 6.4	157.0 ± 6.4	0.561
Weight, kg	59.6 ± 9.3	48.7 ± 8.6	<0.001
BMI, kg/m^2^	24.3 ± 3.3	19.7 ± 2.8	<0.001
Duration of RA, month	103.9 ± 67.8	107.8 ± 66.9	0.676
Erosion, *n* (%)	57 (21.4)	20 (37.0)	0.015
ESR mm/hr	15.5 ± 15.1	16.9 ± 15.1	0.559
CRP mg/dL	0.7 ± 1.2	0.7 ± 1.2	0.837
Visual analogue score	27.2 ± 17.8	26.3 ± 14.6	0.904
Tender joint count	4.0 ± 4.4	3.8 ± 3.3	0.671
Swollen joint count	1.1 ± 2.5	1.2 ± 1.7	0.058
DAS28	3.0 ± 1.2	3.1 ± 1.0	0.669
Current dose of GC, mg	1.3 ± 1.3	1.8 ± 1.3	0.011
Cumulative dose of GC, g	3.3 ± 3.6	5.3 ± 6.0	0.002
MTX, *n* (%)	158 (59.4)	34 (63.0)	0.626
TNF inhibitor, *n* (%)	15 (5.6)	8 (14.8)	0.017
Synthetic DMARDs, *n* (%)	82 (30.8)	23 (42.6%)	0.094
Osteoporosis, *n* (%)	107 (40.2)	34 (63.0)	0.002
Mean strength of grip	19.6 ± 5.6	18.1 ± 6.2	0.038
Walk speed, m/s	0.84 ± 0.16	0.82 ± 0.17	0.418

RA, rheumatoid arthritis; BMI, body mass index; ESR, erythrocyte sedimentation rate; CRP, C-reactive protein; DAS28, Disease Activity Score 28; GC, glucocorticoids; MTX, methotrexate; TNF, tumor necrosis factor; DMARDs, disease modifying anti-rheumatic drugs. Values are mean ± standard deviation or n (%). * Low muscle mass: aLM <5.4 kg/m^2^ female, <7.0 kg/m^2^ male by the European Working Group for Sarcopenia (EWGS).

**Table 3 jcm-10-03458-t003:** Comparison of clinical characteristics between RA patients with sarcopenia and those without by SARC-F.

	No Sarcopenia by SARC-F *n* = 282	Sarcopenia by SARC-F *n* = 38	*p*-Value
Age, year	59.6 ± 9.7	67.3 ± 8.7	<0.001
Male, *n* (%)	20 (7.1)	0	NA
Height, cm	157.4 ± 6.1	152.1 ± 6.8	<0.001
Weight, kg	57.8 ± 10.1	57.8 ± 9.2	0.575
BMI, kg/m^2^	23.3 ± 3.7	24.9 ± 3.5	0.007
Duration of RA, month	104.7 ± 6.4	103.2 ± 69.8	0.867
Erosion, *n* (%)	66 (23.4)	11 (28.9)	0.454
ESR mm/hr	15.4 ± 15.2	18.1 ± 14.1	0.071
CRP mg/dL	0.8 ± 1.7	0.6 ± 1.4	0.348
Visual analogue score	26.0 ± 16.4	35.3 ± 21.3	0.015
Tender joint count	3.5 ± 3.5	7.0 ± 7.0	0.001
Swollen joint count	0.9 ± 1.8	2.9 ± 4.6	<0.001
DAS28	2.9 ± 1.1	3.8 ± 1.4	<0.001
Current dose of GC, mg	1.2 ± 1.0	1.4 ± 1.3	0.011
Cumulative dose of GC, g	3.7 ± 4.2	2.8 ± 3.0	0.002
MTX, *n* (%)	167 (59.2)	25 (65.8)	0.438
TNF inhibitor, *n* (%)	20 (7.1)	3 (7.9)	0.858
Synthetic DMARDs, *n* (%)	96 (34)	10 (26.3)	0.343
Osteoporosis, *n* (%)	120 (42.6)	21 (55.3)	0.139
Total mass index, kg/m^2^	27.5 ± 11.2	22.2 ± 9.9	0.005
Fat free mass index, kg/m^2^	17.9 ± 5.8	19.7 ± 5.3	0.032
aLM, kg/m^2^	7.3 ± 1.1	7.6 ± 1.3	0.036
Walk speed, m/s	0.84 ± 0.16	0.78 ± 0.16	0.035
Mean grip strength, kg	19.8 ± 5.6	15.4 ± 5.4	<0.001

RA, rheumatoid arthritis; SARC-F, sluggishness, assistance in walking, rising from a chair, climb stairs, falls; BMI, body mass index; ESR, erythrocyte sedimentation rate; CRP, C-reactive protein; DAS28, Disease Activity Score 28; GC, glucocorticoids; MTX, methotrexate; TNF, tumor necrosis factor; DMARDs, disease modifying anti-rheumatic drugs; aLM, appendicular lean mass index. Values are mean ± standard deviation or n (%).

**Table 4 jcm-10-03458-t004:** Multiple regression analysis for low muscle mass and sarcopenia by SARC-F in RA patients.

	Low Muscle Mass		Sarcopenia by EWGS		Sarcopenia by AWGS		Sarcopenia by SARC-F	
	OR	*p*-Value	OR	*p*-Value	OR	*p*-Value	OR	*p*-Value
Sex, male	140.65 (20.32–973.54)	<0.001	3.58 (0.71–18.1)	0.12	41.03 (3.41–193.79)	0.003	NA	
Age	1.05 (1.0–1.1)	0.074	1.04 (0.99–1.1)	0.14	1.19 (1.01–1.39)	0.038	1.11 (1.06–1.17)	<0.001
BMI	0.41 (0.31–0.53)	<0.001	0.66 (0.55–0.8)	<0.001	0.47 (0.3–0.75)	0.002	1.13 (1.02–1.24)	0.015
RA duration	-		1.01 (1.0–1.01)	0.09	NA		NA	
Erosion	1.79 (0.72–4.54)	0.247	1.41 (0.51–3.96)	0.51	NA		NA	
DAS28	0.84 (0.58–1.24)	0.844	1.02 (0.64–1.62)	0.93	NA		1.95 (1.38–2.73)	<0.001
Cumulative dose of GC	1.0 (1.0–1.0)	0.437	1.0 (1.0–1.0)	0.36	NA		1.0 (1.0–1.0)	0.48
TNF inhibitor use	4.84 (1.1–21.37)	0.037	NA		NA		NA	

SARC-F, sluggishness, assistance in walking, rising from a chair, climb stairs, falls; RA, rheumatoid arthritis; EWGS, European Working Group for Sarcopenia; AWGS, Asian Working Group for Sarcopenia; OR, odds ratio; BMI, body mass index; DAS28, Disease Activity Score 28; GC, glucocorticoids; TNF, tumor necrosis factor, NA, not applicable

## Data Availability

The data presented in this study are available on request from the corresponding author. The data are not publicly available due to further analyses.

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
