# Peer review of "The Assessment of Muscle Mass and Function in Patients with Long-Standing Rheumatoid Arthritis"

_jcm, 2021, doi:10.3390/jcm10163458_

Round 1

Reviewer 1 Report

Overall it is a well-written manuscript with an interesting topic. However, there are some concerns I have listed below that should be addressed under my opinion.

Point 1. Objective.  Authors have stated in the objective that SARC-F is used as a classification criteria of sarcopenia. However in the references they have used (Malstroms 2016) it is not applied in this sense. Malstroms 2016 as well as Cruz-Jentoff 2019 defined it as valid and consistent for identifying people at risk of sarcopenia associated adverse outcomes. Therefore, it is not applied the same way as the classification criteria of the EWGS and the AWGS. Please explain more clearly the use of SARC-F in the objective.

Point 2. Material and Methods. Inclusion or exclusion criteria could be more broadly explained: was there an age range for participants recruitment? If they suffered any other chronic or acute diseases were they excluded? Recent hospitalisations was an exclusion criteria?

Point 3. Material and Methods. In relation to the medical history, symptoms and physical examination findings, they could be more clearly explained since they appear in the Tables of results, but it is not explained in methods. When were this measurements or questions asked? How were they done? With validated questionnaires or ad-hoc questionnaires? In relation to the VAS it appears in the tables but not in methods, so it is not clear what part of the body it was assessing?

Point 4. Material and Methods. As in a previous comment, in lines 95 and 96, more accurate information should be given. EWGS and AWGS do have classification criteria and can classify as pre-sarcopenia, sarcopenia and severe sarcopenia. But SARC-F is a screening tool. Please explain more accurately.

Point 5. Material and Methods. A flow diagram could help clarify the process that has been followed in order to show how many participants were classified with one or other criteria.

Point 6. Results. This is stated in the abstract “54 patients 16 (16.9%) were determined to have low muscle mass with normal muscle function, as defined by the  EWGS; and 38 patients (11.9%) reported sarcopenia by SARC-F questionnaire.” And this in results: “Nineteen patients (5.9%) were defined as having sarcopenia,”. Please in the results section clarify numbers and criteria classifications.

Point 7. Discussion. Do authors have any explanation to the low rate of men in the sample?

Minor comments related to editing:

  • Line 55: are the references correct? It seems to me it should be number 8 and 14. Please check.
  • Line 65: correct SACR-F
  • Line 127: appendicular, please correct.
  • Line 207: subdivided, please correct.
  • Lines 291 and 292: muslce / quntity

Author Response

Dear Reviewer,

 We appreciate your review of our manuscript “The Assessment of Muscle Mass and Function in Patients with Long-Standing Rheumatoid Arthritis". In response to your comments, we have made several changes and added the necessary clarifications, as summarized below:

Point 1. Objective.  Authors have stated in the objective that SARC-F is used as a classification criteria of sarcopenia. However in the references they have used (Malstroms 2016) it is not applied in this sense. Malstroms 2016 as well as Cruz-Jentoff 2019 defined it as valid and consistent for identifying people at risk of sarcopenia associated adverse outcomes. Therefore, it is not applied the same way as the classification criteria of the EWGS and the AWGS. Please explain more clearly the use of SARC-F in the objective.

Answer) Thank you for the comment. We added it in third phrase of the Introduction as below, and the change is underlined in the revised version of the manuscript.

  • SARC-F has been considered valid and consistent for identifying people at a risk of sarco-penia associated adverse outcomes, and had a consistent reliability and validity in the African American Health study, Baltimore Longitudinal Study of Aging, and National Health and Nutrition Examination Survey.

Point 2. Material and Methods. Inclusion or exclusion criteria could be more broadly explained: was there an age range for participants recruitment? If they suffered any other chronic or acute diseases were they excluded? Recent hospitalisations was an exclusion criteria?

Answer) Thank you for the comment. We added some descriptions including the inclusion or exclusion criteria for study subject, and the change is underlined in the revised version of the manuscript. Recent hospitalization was not an exclusion criteria, but we excluded the patients who suffered serious infection during recent 3 months.

  • All of them were 18 years of age or older and less than 80 years old, diagnosed RA within 10 years, and had no history of malignancy and other autoimmune or inflammatory disorders.

Point 3. Material and Methods. In relation to the medical history, symptoms and physical examination findings, they could be more clearly explained since they appear in the Tables of results, but it is not explained in methods. When were this measurements or questions asked? How were they done? With validated questionnaires or ad-hoc questionnaires? In relation to the VAS it appears in the tables but not in methods, so it is not clear what part of the body it was assessing?

Answer) Thank you for the comment. Data collections were done through chart review and interview, since all participants were enrolled in the clinic of University hospital. Their detailed medical history and initial manifestation with current status could be collected by chart review, but physical examination including tender and swollen joint count, and VAS were conducted by interview for accuracy. VAS is regarded as a pain score in joints due to RA. We added it as below, and the change is underlined in the revised version of the manuscript.

  • Data related to medical history, symptoms, and physical examination findings, laboratory results were collected through chart review and interview, and entered into a database.

Point 4. Material and Methods. As in a previous comment, in lines 95 and 96, more accurate information should be given. EWGS and AWGS do have classification criteria and can classify as pre-sarcopenia, sarcopenia and severe sarcopenia. But SARC-F is a screening tool. Please explain more accurately.

Answer) Thank you for the comment. We described more clearly as your recommendation, and the change is underlined in the revised version of the manuscript.

  • Having low muscle mass without an impairment on muscle strength or physical perfor-mance was defined as “presarcpenia”, and “sarcopenia” is defined as low muscle mass with low muscle strength or low physical performance. The SARC-F is a screening ques-tionnaire, consisting of five questions, pertaining to muscle strength, walking aids, getting up from a chair, climbing stairs, and falling

Point 5. Material and Methods. A flow diagram could help clarify the process that has been followed in order to show how many participants were classified with one or other criteria.

Answer) Thank you for the comment. We made a flow diagram as Supplement Figure 1, but most of the patients, who were suspected to have sarcopenia by SARC-F (n = 34), were not included presarcopenia or sarcopenia defined as EWSG or AWGS. 

Point 6. Results. This is stated in the abstract “54 patients 16 (16.9%) were determined to have low muscle mass with normal muscle function, as defined by the  EWGS; and 38 patients (11.9%) reported sarcopenia by SARC-F questionnaire.” And this in results: “Nineteen patients (5.9%) were defined as having sarcopenia,”. Please in the results section clarify numbers and criteria classifications.

Answer) Thank you for the comment. We corrected them as below, and the change is underlined in the revised version of the manuscript.

  • Nineteen patients (5.9%) were defined as having sarcopenia as defined by the EWGS, and 54 patients (16.9%) had low muslce mass without low muscle strength or physical performance (presarcopena) as defined by the EWGS. And, 7 patients (2.2%) were defined as having sarcopenia as defined by the AWGS, 38 patients (11.9%) were reported sarcopenia by SARC-F questionnaire.

Point 7. Discussion. Do authors have any explanation to the low rate of men in the sample?

Answer) Thank you for your decisive opinion. We considered and discussed such points, and concluded there were some cases which had a difficulties to enroll male patients with RA. We explained them as below in Discussion, and the change is underlined in the revised version of the manuscript.

  • In addition, the number of male patients was small compared to the general proportion of male patients with RA. It is considered that there were many cases in which male patients were reluctant to admit that they lacked muscle or muscle strength even in the hospital, or were reluctant to provide sufficient time for joining investigations. Despite all subjects were enrolled equally according to the established criteria, there might be a selection bias in this study.

Minor comments related to editing:

  • Line 55: are the references correct? It seems to me it should be number 8 and 14. Please check.
  • Line 65: correct SACR-F
  • Line 127: appendicular, please correct.
  • Line 207: subdivided, please correct.
  • Lines 291 and 292: muslce / quntity

Answer) Thank you for your detailed comments, we corrected them, and the change is underlined in the revised version of the manuscript.

We thank you for the constructive review and hope that the revised manuscript now meets the journal's standards for publication.

Reviewer 2 Report

The manuscript titled, "The Assessment of Muscle Mass and Function in Patients with 2 Long-Standing Rheumatoid Arthritis, reports different groups identified by the sarcopenia classification criteria of the EWGS, AWGS, and SACR-F, and determine the clinical features associated with low muscle mass or strength in Korean patients with RA.

The manuscript is written well and can be published with minor corrections. I do not see any significant novelty in the work but still, the data provides useful information about the assessment of Muscle Mass and Function in Patients with 2 Long-Standing RA.

Author Response

Dear Reviewer,

We appreciate your review of our manuscript “The Assessment of Muscle Mass and Function in Patients with Long-Standing Rheumatoid Arthritis". In response to your comments, we have made several changes and added the necessary clarifications, as summarized below:

The manuscript titled, "The Assessment of Muscle Mass and Function in Patients with Long-Standing Rheumatoid Arthritis, reports different groups identified by the sarcopenia classification criteria of the EWGS, AWGS, and SACR-F, and determine the clinical features associated with low muscle mass or strength in Korean patients with RA.

The manuscript is written well and can be published with minor corrections. I do not see any significant novelty in the work but still, the data provides useful information about the assessment of Muscle Mass and Function in Patients with Long-Standing RA.

Answer) Thank you for the comment. We corrected some sentences, and the change is underlined in the revised version of the manuscript.

We thank you for the constructive review and hope that the revised manuscript now meets the journal's standards for publication.

Round 2

Reviewer 1 Report

I have reviewed the manuscript, I can see the authors have applied changes related to previous comments. I think the manuscript has been strengthen and I do not have more comments.